# Endoglin Is an Important Mediator in the Final Common Pathway of Chronic Kidney Disease to End-Stage Renal Disease

**DOI:** 10.3390/ijms24010646

**Published:** 2022-12-30

**Authors:** Tessa Gerrits, Isabella J. Brouwer, Kyra L. Dijkstra, Ron Wolterbeek, Jan A. Bruijn, Marion Scharpfenecker, Hans J. Baelde

**Affiliations:** 1Department of Pathology, Leiden University Medical Centre, 2333 ZA Leiden, The Netherlands; 2Department of Biomedical Data Sciences, Medical Statistics, Leiden University Medical Centre, 2333 ZA Leiden, The Netherlands

**Keywords:** endoglin, fibroblast, interstitial fibrosis, TGF-β

## Abstract

Chronic kidney disease (CKD) is a slow-developing, progressive deterioration of renal function. The final common pathway in the pathophysiology of CKD involves glomerular sclerosis, tubular atrophy and interstitial fibrosis. Transforming growth factor-beta (TGF-β) stimulates the differentiation of fibroblasts towards myofibroblasts and the production of extracellular matrix (ECM) molecules, and thereby interstitial fibrosis. It has been shown that endoglin (*ENG*, CD105), primarily expressed in endothelial cells and fibroblasts, can function as a co-receptor of TGF signaling. In several human organs, endoglin tends to be upregulated when chronic damage and fibrosis is present. We hypothesize that endoglin is upregulated in renal interstitial fibrosis and plays a role in the progression of CKD. We first measured renal endoglin expression in biopsy samples obtained from patients with different types of CKD, i.e., IgA nephropathy, focal segmental glomerulosclerosis (FSGS), diabetic nephropathy (DN) and patients with chronic allograft dysfunction (CAD). We showed that endoglin is upregulated in CAD patients (*p* < 0.001) and patients with DN (*p* < 0.05), compared to control kidneys. Furthermore, the amount of interstitial endoglin expression correlated with eGFR (*p* < 0.001) and the amount of interstitial fibrosis (*p* < 0.001), independent of the diagnosis of the biopsies. Finally, we investigated in vitro the effect of endoglin overexpression in TGF-β stimulated human kidney fibroblasts. Overexpression of endoglin resulted in an enhanced *ACTA2*, *CCN2* and *SERPINE1* mRNA response (*p* < 0.05). It also increased the mRNA and protein upregulation of the ECM components collagen type I (*COL1A1*) and fibronectin (*FN1*) (*p* < 0.05). Our results suggest that endoglin is an important mediator in the final common pathway of CKD and could be used as a possible new therapeutic target to counteract the progression towards end-stage renal disease (ESRD).

## 1. Introduction

Chronic kidney disease (CKD) is a slow-developing, progressive deterioration of renal function, that can ultimately progress towards end-stage renal disease (ESRD). Globally, 1.2 million people died in 2017 from CKD and its complications, and the prevalence of CKD is still rising [1]. CKD may result from various causes of renal dysfunction of sufficient magnitude, including diabetic nephropathy (DN) and different glomerulopathies such as IgA nephropathy or focal segmental glomerulosclerosis (FSGS) [2,3]. Clinically, CKD is characterized by high blood pressure, anemia, fatigue, itching, nausea and muscle cramps, with the severity ranging from mild symptoms to ESRD. Patients with ESRD have an estimated glomerular filtration rate (eGFR) below 15 mL/min/1.73 mL, meaning that these patients become dialysis dependent, or become in need of a kidney transplantation [4]. Histologically, CKD is characterized by glomerular sclerosis, tubular atrophy, interstitial infiltration and interstitial fibrosis, irrespective of the type of renal disease [5].

Interstitial fibrosis is considered to be an excessive accumulation of extracellular matrix (ECM), especially of collagen type I (*COL1A1*), collagen type III and fibronectin (*FN1*) [6]. When interstitial fibrosis develops, fibroblasts increase in number and can differentiate into myofibroblasts, which are responsible for the synthesis and accumulation of interstitial ECM components [7]. Myofibroblasts are characterized by abundant expression of alpha-smooth muscle actin (α-SMA; *ACTA2*) [8]. Transforming growth factor-beta (TGF-β) is an important cytokine in the differentiation of fibroblasts towards myofibroblasts and the production of ECM molecules [9,10]. There are three isoforms of TGF-β in mammals; TGF-β1, TGF-β2 and TGF-β3 [11]. TGF-β1 is considered a profibrotic mediator in various kidney diseases [12]. In canonical TGF-β signaling, active TGF-β1 binds to type II TGF-β receptor (TβRII), which recruits and activates type I TGF-β receptor (TβRI) and downstream receptor-associated Smads, i.e., Smad2 and Smad3. Phosphorylated Smad2/3 forms an oligomeric complex with Smad4 and subsequently translocates into the nucleus to regulate transcription of target genes, such as connective tissue growth factor (*CCN2*) and plasminogen activator inhibitor-1 (*SERPINE1*) [13]. Additionally, the non-canonical TGF-β signaling pathways can play a role in the development of renal fibrosis, through activation of JNK/p38, ERK, PI3K/Akt or JAK2/STAT3 signaling.

It has been shown that the homodimeric transmembrane glycoprotein endoglin (*ENG*, CD105) [14], can function as a co-receptor of the TβRI-TβRII complex [15,16]. Endoglin is mainly known for its role in angiogenesis [17,18,19,20]. More recently, more research on the role of endoglin in fibrosis is being conducted, as summarized by Schoonderwoerd et al. (2020) [21]. In several human organs, endoglin tends to be upregulated when chronic damage and fibrosis is present. In humans, it has been described that endoglin is upregulated in livers with chronic injury in hepatic stellate cells, liver sinusoidal endothelial cells and Kupffer cells [22]. In human subjects with heart failure, endoglin expression is increased in cardiac fibroblasts of the failing left ventricle [23]. Additionally, in patients with scleroderma, endoglin was found to be significantly upregulated in lesional systemic sclerosis fibroblasts [24]. In the kidney, it has been shown that in several animal models of renal fibrosis, including unilateral ureter obstruction (UUO), ischemia-reperfusion injury, and radiation-induced nephropathy, endoglin is upregulated in the renal interstitium [25,26,27]. In line with these findings, using the UUO model, in endoglin overexpressing mice fibrosis is aggravated [28]. Accordingly, endoglin haploinsufficient mice have reduced radiation-induced renal fibrosis, as well as reduced numbers of myofibroblasts compared to wild-type mice [29]. In 1997, Roy-Chaudhury et al. investigated kidneys in a small cohort of diseased patients and found that endoglin is upregulated in a variety of kidney diseases [30]. Later, we have shown that, in patients with DN, endoglin is upregulated in myofibroblasts of the renal interstitium and that this correlates with interstitial fibrosis and clinical renal outcome [31].

Since interstitial fibrosis is one of the major hallmarks of the final common pathway to ESRD, we hypothesized that endoglin is upregulated in interstitial fibrosis regardless of the primary renal disease and contributes to the progression to ESRD. In this study, we measured renal endoglin expression in biopsy samples obtained from patients with different types of CKD, i.e., IgA nephropathy, FSGS, DN and patients with chronic allograft dysfunction (CAD). We correlated these results with the degree of interstitial fibrosis and renal function. In addition, we investigated the effect of endoglin overexpression in human kidney fibroblasts in vitro on the TGF-β induced pro-fibrotic signaling. Our results suggest that endoglin is an important mediator in the final common pathway of CKD and could be used as a possible new therapeutic target to counteract the development of renal fibrosis and progressive decline of renal function in CKD.

## 2. Results

### 2.1. Endoglin Expression Is Increased in Various Chronic Kidney Diseases and Correlates with Renal Function and the Amount of Interstitial Fibrosis

First, we stained paraffin-embedded serial sections of renal biopsies from patients with CAD, DN, FSGS and IgA nephropathy for endoglin and Sirius Red, which indicates the presence of collagen and measured the positively stained areas in the interstitium. As a control, the renal tissue of kidneys excluded for transplantation (ET) was stained. Representative images of the interstitial endoglin and Sirius Red staining of a control kidney and of different renal diseases are depicted in Figure 1A–J. Endoglin-positive staining was observed in the same areas as Sirius Red staining. The expression of endoglin was significantly higher in patients with CAD and DN compared to ET controls (respectively *p* < 0.001; *p* < 0.05; Figure 1K). The highest percentage of Sirius Red-positive area compared to ET controls was seen in patients with CAD, DN and FSGS (respectively *p* < 0.001; *p* < 0.001 *p* < 0.01; Figure 1L).

To investigate the relation between endoglin expression and renal function, we correlated the endoglin-positive interstitial area with the eGFR. First, we performed a regression analysis for all patient groups separately. We then analyzed the interaction between the different kidney diseases and endoglin expression to be able to compare the angles of inclination. Since the angle of inclination of all the groups did not differ significantly (*p* = 0.85), we considered it justified to calculate a pooled (common) regression line for all the patient groups together. When CAD, DN, FSGS and IgA nephropathy were pooled, the endoglin-positive interstitial area had a significant inverse correlation with the eGFR (*p* < 0.001; Figure 2A). The regression line had an R^2^ of 0.24, meaning that patients with the more endoglin-positive interstitial area have a lower eGFR on average. Subsequently, we correlated the endoglin-positive interstitial area with the Sirius Red-positive interstitial area in a similar way, with a correction applied for eGFR. In CAD and IgA nephropathy patients, endoglin and Sirius red stained areas were positively correlated (*p*-value of, respectively 0.05 and 0.03). The correlation found between endoglin and Sirius Red levels in DN and FSGS patients was not significant. Again, we analyzed the interaction for every regression line. As the angle of inclination of all the groups did not differ significantly (*p* = 0.72), we also analyzed the patient groups pooled together. This showed that the endoglin-positive interstitial area had a strong significant correlation with the Sirius Red-positive interstitial area (*p* < 0.001; Figure 2A), with an R^2^ of 0.32, indicating that the extent of Sirius Red positivity is on average more excessive when a more endoglin-positive interstitial area is present in a biopsy.

### 2.2. Endoglin Overexpression Increases the TGF-β-Induced Profibrotic Response in Human Kidney Fibroblast

Next, we studied the TGF-β profibrotic response in endoglin overexpressing TK173 (human kidney fibroblast) cells in vitro. Cells were transduced with a lentiviral construct containing the sequence for full-length *ENG* mRNA (Endoglin knock-in cells; *ENG^KI^*); control cells were transduced with a virus expressing empty vector control (*ENG^ctrl^*). Both the endoglin construct and the control construct contained a Green Fluorescent Protein (*GFP*) sequence to be able to measure construct uptake. Transduction caused a 20-fold increase in endoglin expression at the mRNA level in *ENG^KI^* compared to *ENG^ctrl^* (Figure 3A). *GFP* mRNA levels were not significantly different between the two transduced cell lines (not shown). The increase in endoglin mRNA expression resulted in a 9-fold increase in endoglin protein levels in *ENG^KI^* compared to *ENG^ctrl^* (*p* < 0.001; Figure 3B,C). The *ENG* mRNA and endoglin protein levels in the *ENG^ctrl^* were comparable to TK173, without significant differences (Figure 3B,C).

To investigate the effect of the increased endoglin expression on the TGF-β induced fibrotic response, *ENG^KI^* and *ENG^ctrl^* fibroblasts were stimulated with 5 ng/mL TGF-β1, and mRNA expression of *ACTA2*, *CCN2*, *SERPINE1*, *FN1* and *Col1A1* was measured using quantitative real-time PCR (qPCR). Endoglin overexpression significantly increased the TGF-β1 induced mRNA expression of *ACTA2* (*p* < 0.05; Figure 4A). Additionally, the fold change in *CCN2* and *SERIPINE1* mRNA expression levels were significantly increased in *ENG^KI^* compared to *ENG^ctrl^* (*p* < 0.01 and *p* < 0.001, respectively; Figure 4B,C). Furthermore, the TGF-β1-mediated fold increase in *COL1A1* mRNA and *FN1* mRNA were significantly higher in *ENG^KI^* compared to *ENG^ctrl^* (*p* < 0.05 and *p* < 0.05, respectively; Figure 4D,E).

Finally, a Western blot was performed to measure the protein levels of fibronectin and collagen type I as a readout for the final result of pro-fibrotic TGF-β1 signaling. The TGF-β1-induced response was augmented in *ENG^KI^* compared to *ENG^ctrl^* for both fibronectin (*p* < 0.05; Figure 5A,C) and collagen type I (*p* < 0.05; Figure 5B,D) which confirms the results at the mRNA level.

## 3. Discussion

Here, we show that endoglin is upregulated in the interstitium of patients with CKD compared to ET control kidneys. Endoglin was present in the areas where Sirius Red was most prominent. Furthermore, the interstitial area positive for endoglin correlated with the eGFR; this finding was independent of the patient’s diagnosis. These results indicate that endoglin is upregulated in the final common pathway of CKD rather than being of relevance in one specific kidney disease. Importantly, we show that overexpression of endoglin increases the profibrotic response in TGF-β1-stimulated human kidney fibroblasts. This effect was shown at the mRNA level for *ACTA2*, a precursor of the differentiation marker α-SMA [32], for *CCN2* and *SERPINE1*, both pro-fibrotic downstream targets of the TGF-β1 signaling pathway [33,34] and for *COL1A1* and *FN1*, which translated into an increase in protein expression of the ECM molecules collagen type I and fibronectin [6]. Together, these results indicate that endoglin is not only overexpressed in patients with progressive chronic kidney disease but also suggest that overexpression of endoglin in the interstitium contributes to interstitial fibrosis.

Our finding that endoglin is upregulated in various chronic kidney diseases is in line with current literature. For instance, it has been described that endoglin is upregulated in cardiac fibroblasts in the left ventricle of patients with heart failure [23]. Upregulation of endoglin has also been described in vascular smooth muscle cells in human atherosclerotic plaques of the aorta [35]. Moreover, endoglin upregulation has been reported in fibroblasts in cutaneous skin lesions in patients with scleroderma [24], in human hepatic stellate cells [22] and in fibroblasts isolated from strictures in Crohn’s disease [36]. Finally, we previously showed that endoglin is upregulated in myofibroblasts in the interstitium of patients with diabetic nephropathy [31].

The first evidence of a possible link between endoglin in the kidney interstitium and the extent of chronic histological renal damage in different kidney diseases was investigated by Roy-Chaudhury et al. This study from 1997 analyzed biopsies of patients with chronic progressive renal disease and found a weak, albeit significant difference between the extent of interstitial endoglin staining in cases with mild and severe chronic histological damage [30]. However, patient cohorts were relatively small and results were scored semi-quantitatively. By using larger cohorts and quantitative image analyses, we were able to demonstrate a significant increase in interstitial endoglin in patients with chronic kidney disease, which was accompanied by higher Sirius Red staining, indicative of increased collagen deposition. Even more important, we were able to show that an increase in the endoglin-positive interstitial area was significantly correlated with a decrease in eGFR and that this finding was independent of the diagnosis. Pooling of all individual measurements of the different patient cohorts in one analysis was considered justified, as the interaction term for the various slopes of the regression lines of endoglin with Sirius Red or eGFR did not differ significantly between the cohorts.

One plausible explanation why interstitial endoglin and eGFR only correlated when combining all patient cohorts could be that patient numbers in the individual groups were too small and our study therefore underpowered. Evidence for this comes from the two largest cohorts, scilicet IgA nephropathy and CAD, which also individually showed a significant correlation between the endoglin and Sirius Red-positive area. Taken together, these findings support our hypothesis that endoglin is upregulated in the interstitium of patients with CKD and relates to the amount of interstitial fibrosis and a decline in renal function.

To further explore the role of endoglin in the fibrotic interstitium and the consequences of being overexpressed, we performed in vitro experiments with endoglin overexpressing human kidney fibroblasts. We found that endoglin contributes to the TGF-β-mediated excessive production of ECM proteins in human renal fibroblasts. This is different from other in vitro studies investigating endoglin in hepatic stellate cells [22] and intestinal fibroblasts [36], where they conclude that endoglin is only upregulated as a compensatory mechanism, not contributing to the development of fibrosis. Yet, several studies in in vivo models predominantly show a profibrotic function for endoglin in the kidney. For example, Oujo et al. reported that transgenic mice overexpressing human endoglin have increased renal fibrosis following UUO [26]. Additionally, Docherty et al. described that renal ischemia-reperfusion has a less severe effect in heterozygous *ENG*-knockout (*ENG*^+/−)^ mice compared to wild-type mice manifesting in less acute tubular necrosis and less increase in creatinine levels [23]. Moreover, in a radiation injury model of chronic renal fibrosis, *ENG*^+/−^ mice developed less macrophage infiltration, fibrosis and have improved renal function compared to wild-type littermates [29,37,38]. Furthermore, we previously showed that a reduction in endoglin in cultured human renal fibroblasts attenuates *ACTA2*, *CCN2* and *SERPINE1* mRNA expression, indicating that reducing endoglin could be a potential therapeutic strategy in the future [31]. Taken together, the results of the in vivo studies, our previous study and the result shown here strongly suggest that endoglin works as a pro-fibrotic factor in the kidney.

We show that endoglin, the co-receptor of the TβR is upregulated in the human interstitium in CKD. TβR-I/II/III are also upregulated in tubulointerstitial lesions in CKD with increased matrix deposition, such as FSGS and IgA nephropathy [39]. Moreover, the expression of the ligand TGF-β is increased in the tubulointerstitium in CKD, such as CAD [40,41]. In addition, we show that overexpression of endoglin in kidney fibroblast increases the TGF-β-mediated upregulation of pro-fibrotic downstream targets of TGF-β, resulting in excessive ECM production. This is of importance, as TGF-β is already overexpressed in CKD and therefore additional endoglin upregulation could make interstitial fibrosis and renal function decline even more pronounced.

It would be interesting to explore if endoglin could be a therapeutic target in chronic kidney disease. During the progression of CKD to ESRD, the fibrosis is part of a vicious circle in which the fibrosis leads to decreased oxygenation of the surrounding tissue, which leads to an upregulation of reactive oxygen species and attraction of inflammatory cells, in its turn again exacerbating fibrosis, independent of the primary renal disease [42]. TGF-β1 is involved in this process, described as a pro-fibrotic factor that also affects the inflammatory response. By intervening within this vicious circle, further aggravation of fibrosis could be prevented. Based on our findings, endoglin could be a suited target for this approach as it acts as a co-receptor of the TβR and aggravates the TGF-β-mediated pro-fibrotic response. When more severe, at a certain, yet undefined stage, fibrosis becomes self-sustainable by positive feedback loops and epigenetic changes, with a point of no return as result [43,44]. It is desirable to intervene before this self-sustainable stage of fibrosis is reached.

Endoglin Targeted Therapy is not new, as it has already been tested as a vascular-targeting antiangiogenic therapy. In cancer therapy, multiple research antibodies against endoglin have been developed, including TRC105 [45]. In 2012, the first human trial showed that treatment with TRC105 was safe, with indications of clinical activity [46], although in following trials, not enough clinical benefit was shown to warrant further clinical development in terms of cancer research [47]. Less research has been conducted in terms of endoglin as a target for anti-fibrotic therapy. In vitro, siRNAs have been successfully used to silence endoglin expression in cardiac and in renal fibroblasts, leading to reduced collagen synthesis [23,31]. Targeting endoglin in cancer-associated fibroblasts by TRC105 inhibited fibroblast invasion in vitro [48]. Knockdown of endoglin by tail vein injection of AAV9-*ENG*, an adeno-associated virus used for gene therapy, meliorated peritoneal thickening and collagen deposition in a mouse model for peritoneal fibrosis [49].

A limitation of our study was that the patient study is descriptive. However, we did find a strong correlation between endoglin and renal function and interstitial fibrosis. Additionally, we only had access to the clinical data available in the pathology reports. This means that we were missing information on a possible decline of renal function or use of medication and that possible confounding factors were not taken along. A prospective follow-up study would be needed to overcome this limitation in the future.

In conclusion, we report that endoglin is upregulated in various CKD, where high interstitial levels of endoglin correspond with a low eGFR and more extensive renal interstitial fibrosis. Furthermore, we have shown in cultured fibroblast that endoglin enhances the pro-fibrotic effects of TGF-β, with as a result enhancing the ECM production. These findings imply an important role of endoglin in the final common pathway to end stage renal disease. Future studies are needed to show whether endoglin can serve as a therapeutic target for reducing the formation of renal fibrosis in patients with chronic kidney disease, with the purpose of slowing the renal decline towards ESRD.

## 4. Materials and Methods

### 4.1. Patient Cohorts

Renal biopsy samples from patients biopsied between 1987 and 2016 were obtained from the pathology archives of the LUMC. In total, 43 renal biopsy samples from kidney transplantation patients biopsied at least six months after transplantation (CAD), 11 biopsy samples from patients with histologically confirmed DN, 25 renal biopsy samples from patients diagnosed with FSGS and 84 biopsies from patients diagnosed with IgA nephropathy were collected. As a control group, we included 7 renal tissue samples of kidneys which were intended to be transplanted but that were eventually excluded for transplantation (ET). Reasons for the exclusion for transplantation were atherosclerosis, a malignant tumor elsewhere in the body or a prolonged ischemic time. The clinical data retrieved from the patients’ pathology reports are shown in Table 1. All patient samples were collected and handled in accordance with Dutch national ethics guidelines (the Code of Conduct for the Proper Secondary Use of Human Tissue).

### 4.2. Histology, Immunohistochemistry and Analysis of Sections

Serial sections (4-µm thick) of paraffin-embedded kidney tissues were subjected to heat-induced antigen retrieval using 10 mM Tris/EDTA (pH 9.0). One section per patient was stained with a goat anti-human endoglin antibody (1:800; R&D Systems, Minneapolis, MN, USA). A consecutive section was stained with Picro-Sirius Red (Sigma-Aldrich, Saint Louis, MO, USA). The kidney sections were digitalized using a Philips Ultra-Fast Scanner 1.6 RA. The renal cortex of the endoglin and Sirius red-stained slides were analyzed using ImageJ 1.48v (National institutes of Health, Bethesda, MD, USA). Glomeruli and large blood vessels were excluded from the analysis. The total positively stained area was measured and divided by the total interstitial area.

### 4.3. Cell Lines and Cell Culture

The human kidney fibroblast cell line TK173 was a kind gift from Prof. Gerhard A. Müller, Department of Nephrology and Rheumatology, Georg August University, Göttingen, Germany [50]. Fibroblasts were transduced as described in the following chapter and cultured in DMEM/F12 (Gibco Laboratories, Gaithersburg, MD, USA) supplemented with 10% fetal bovine serum (Sigma-Aldrich), and 0.4% penicillin-streptomycin (Gibco Laboratories) at 37 °C in 5% CO_2_.

### 4.4. Lentiviral Transduction of TK173 Fibroblasts

Fibroblasts were transduced with a lentiviral vector expressing either *ENG* full-length RNA (hereafter referred to as *ENG^KI^* fibroblasts) or a non-targeting vector control (hereafter referred to as *ENG^ctrl^* fibroblasts). In brief, the full-length *ENG* gene from the pMD18-T endoglin plasmid (pMD-CD105, HG10149-M, Sino Biological Inc., Wayne, PA, USA) was cloned in the pUltra lentiviral transfer vector (plasmid #24129, Addgene, Watertown, MA, USA) [51] and transfected into competent HB101 *E. coli*, which were cultured on two agarose Petri dishes containing 1% ampicillin at 37 °C overnight [52]. Colonies were screened for the inclusion of a plasmid containing the endoglin gene by qPCR. Positive clones ware cultured overnight in 450 mL LB medium containing ampicillin.

HEK293T cells were transfected with a mixture of viral plasmids (Addgene); pCMV-VSV-G [53], pMDLg-RRE (gag/pol) [54], and pRSV-REV [54], and the pUltra plasmid containing *ENG*. Transfection efficiency was assessed by fluorescence microscopy with the EVOS FLoid Cell Imaging Station (Thermo Fisher Scientific, Waltham, MA, USA). 48 and 72 h after transfection, lentiviruses were harvested from the supernatant and filtered through a sterile low-protein binding 0.45 µm Whatman™ filter (GE Healthcare, Chicago, IL, USA). A 1:500 dilution of the medium, with lentiviruses in the medium without penicillin-streptomycin was prepared and transduced into TK173 human kidney fibroblasts. The transduced cells were then seeded in a 96-wells plate for limiting dilution with a concentration 0.8 cells per well to obtain monoclonal colonies.

### 4.5. Analysis of Transduction Efficiency

To determine the efficiency of *ENG* knock-in in the clones, total RNA was isolated from *ENG^ctrl^* and *ENG^KI^* fibroblasts one day after reaching confluence using TRIzol (Ambion, Foster City, CA, USA) in accordance with the manufacturer’s instructions. The RNA was converted to cDNA, which was analyzed with qPCR for *ENG* and *GFP* using the IQTM SYBR Green Supermix (Bio-Rad, Hercules, CA, USA) with a CFX real-time system (Bio-Rad). Cycle threshold (Ct) values were normalized to the housekeeping gene Hypoxanthine phosphoribosyltransferase 1 (*HPRT1*). The primers used are shown in Table 2. One colony transduced with the vector containing endoglin and one colony transduced with the empty vector were selected for further experiments based on morphology, comparable growth speed, endoglin overexpression and comparable expression of *GFP*.

For analysis with Western blotting, one day after reaching confluence, *ENG^ctrl^* and *ENG^KI^* fibroblasts were washed in ice-cold phosphate-buffered saline (PBS) and lysed in Tris-buffered saline (TBS) (pH 7.5) containing 1% (*w*/*v*) sodium dodecyl sulfate (SDS), 10 mM EDTA, and protease and phosphatase inhibitor cocktails (Roche, Basel, Switzerland). Protein concentration was determined using a detergent-compatible protein assay (Bio-Rad). Protein lysates were separated by SDS-PAGE and transferred to nitrocellulose membranes for Western blot analysis. The membranes were blocked for 1 h in TBS containing 5% (*w*/*v*) bovine serum albumin (BSA), and then incubated overnight with a primary antibody against endoglin (1:1000, R&D Systems); GAPDH (1:5000, Cell Signaling Technology, Danvers, MA, USA) was used as a loading control. IRDye Infrared Fluorescent antibodies (LI-COR Biosciences, Lincoln, NE, USA) were used as the secondary antibody, and the signals were visualized using the LI-COR Odyssey Infrared Imaging System (LI-COR Biosciences). Band intensity of the endoglin Western blot was quantified using Image StudioTM Lite Software, Version 5.2 (LI-COR Biosciences). Each experiment was performed in triplicate.

### 4.6. Analysis of TGF-β-Induced Changes in Gene and Protein Expression

To analyze the effect of TGF-β1 stimulation on downstream signaling targets, 10^5^
*ENG^KI^* or *ENG^ctrl^* fibroblasts per well were plated in a 12-well plate. One day after the fibroblasts reached confluence, they were further cultured in serum-free medium for five hours and subsequently incubated with 5 ng/mL TGF-β1 (PeproTech) for 24 h. As a control, cells were incubated with 0.1% (*w*/*v*) BSA in PBS without TGF-β1. RNA isolation, cDNA synthesis, and qPCR were performed as described above, and Ct values were normalized to the housekeeping gene *HPRT1*. The primers used are shown in Table 2.

Samples were also subjected to Western blot analysis which was performed as described above. To clear the lysates, the samples were passed through a QIAshredder homogenizer (QIAGEN, Hilden, Germany). As primary antibodies, rabbit anti-collagen-type I primary antibody (IgG, 1:10.000; Abcam, Cambridge, UK), rabbit anti-fibronectin primary antibody (1:1200; Sigma Aldrich, Saint Louis, MO, USA), and Mouse-anti-α-tubulin (1:10.000; Cell Signaling Technology) were used. IRDye Infrared Fluorescent antibodies (LI-COR Biosciences, Lincoln, NE, USA) were used as the secondary antibody. Each experiment was performed in triplicate. Band intensity was quantified using Image StudioTM Lite Software, Version 5.2 (LI-COR Biosciences).

### 4.7. Statistical Analysis

The interstitial endoglin-positive area in ET kidneys was compared to the different CKD groups using student’s *t*-tests derived from a one-way ANOVA after an arcsine transformation. The arcsine transformation is one of the standard transformations for percentages and proportions and takes into account a potential difference in variances [55,56]. This analysis is also applied for the interstitial Sirius Red-positive area. For the correlation between endoglin and eGFR, a univariate regression model was used. For the correlation between the endoglin-positive interstitial area and the Sirius Red-positive interstitial area a univariate regression model was used, whereby eGFR was also implemented in the model to correct for potential confounding. Before formulating the final regression model, an extra analysis was performed with an interaction term between endoglin and the diagnosis included in the model to examine whether the angle of inclination of the individual groups differed significantly. When the hypothesis of equal angles of inclination is not rejected, there are no weighty objections to fit a single regression line for all the groups pooled together. The outcome of cell culture experiments was log-transformed before analyzed by using the Student’s *t*-test, derived from a one-way ANOVA where indicated in the figure legend. A *p*-value < 0.05 was considered statistically significant.

## Figures and Tables

**Figure 1 ijms-24-00646-f001:**
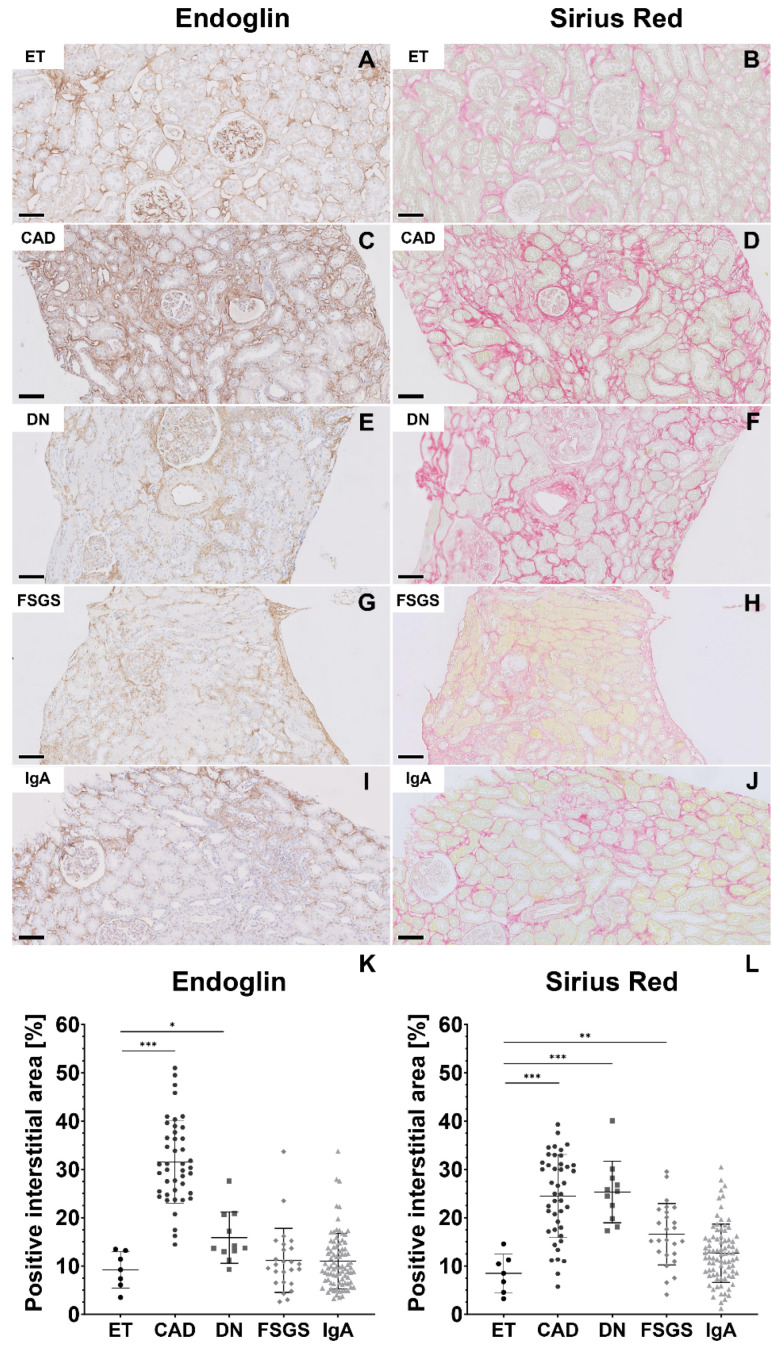
**Endoglin expression is increased in various chronic kidney diseases.** Representative images of endoglin staining (**A**,**C**,**E**,**G**,**I**) and Sirius Red staining (**B**,**D**,**F**,**H**,**J**) in kidney biopsies. Control group consisting of kidney’s excluded for transplantation (**A**,**B**), patients with chronic allograft dysfunction (CAD) (**C**,**D**), patients with histologically confirmed diabetic nephropathy (DN) (**E**,**F**), patients with focal segmental glomerulosclerosis (FSGS) (**G**,**H**) and patients with IgA nephropathy (**I**,**J**). Quantitative analysis of the endoglin-positive interstitial area (**K**) and Sirius Red–positive interstitial area (**L**). * *p* < 0.05, ** *p* < 0.01 and *** *p* < 0.001 (Student’s *t*-tests derived from a one-way ANOVA after arcsine-transformation of the outcome variable). The scale bars represent 100 µm.

**Figure 2 ijms-24-00646-f002:**
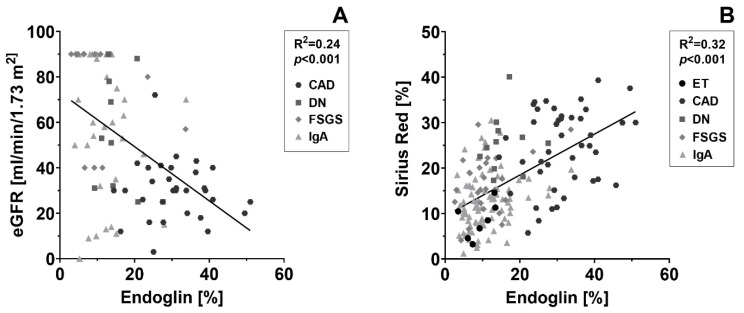
**Endoglin expression correlates with kidney function and interstitial collagen deposition.** (**A**) The interstitial area positive for endoglin is plotted against the estimated glomerular filtration rate (eGFR) per patient. The regression line is based on the pooled measurement of CAD, DN, FSGS and IgA nephropathy, which was considered justified after performing interaction analysis. R^2^ of the regression is 0.24, *p* < 0.001. (**B**) The interstitial area positive for endoglin is plotted against the Sirius Red-positive interstitial area, which is a measure for interstitial fibrosis. The regression line is based on the pooled measurement of ET, CAD, DN, FSGS and IgA nephropathy, which was considered justified after performing interaction analysis. Data were corrected for eGFR, R^2^ of the regression is 0.32, *p* < 0.001.

**Figure 3 ijms-24-00646-f003:**
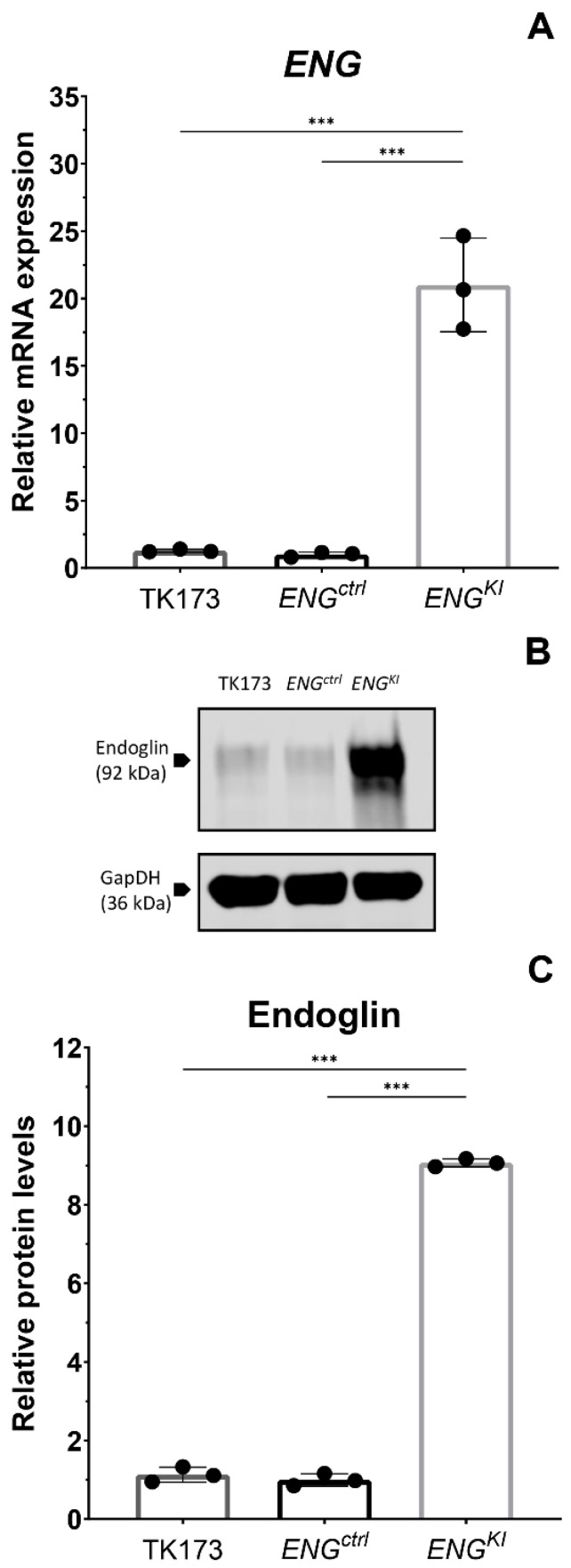
***ENG* mRNA and endoglin protein levels in TK173 human kidney fibroblasts lentivirally transduced with an endoglin and a control construct.** (**A**) *ENG* mRNA levels measured using qPCR in non-transduced TK173 cells and TK173 cells transduced with an empty vector control (*ENG^ctrl^*) or with a vector containing a full length *ENG* RNA sequence (*ENG^K^^I^*). Data are expressed relative to *ENG^ctrl^*. Results were corrected for the housekeeping gene *HPRT*. (**B**) Endoglin protein levels in transduced *ENG^K^*^I^ and *ENG^ctrl^* and non-transduced TK173 cells were determined by Western blotting. The figure depicts a representative Western blot. (**C**) Quantification of the results obtained from the Western blots. Protein levels were corrected for the loading control GAPDH. Endoglin protein is expressed relative to *ENG^ctrl^*. Data are presented as means ± SD from three independent experiments. *** *p* < 0.001 (Student’s *t*-test derived from a one-way ANOVA after log-transformation of the outcome variable).

**Figure 4 ijms-24-00646-f004:**
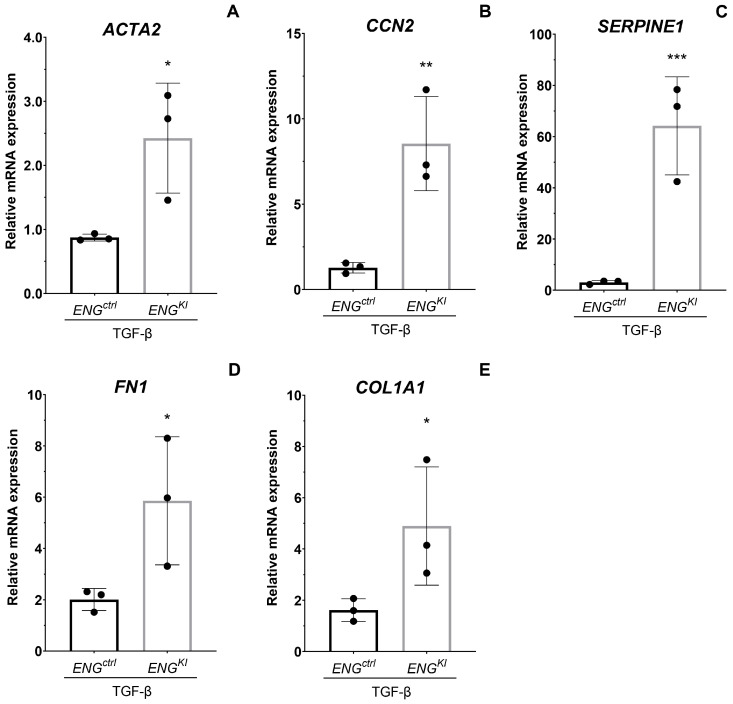
***ENG^K^*^I^ fibroblasts have an increased TGF-β—induced profibrotic response at mRNA level.** *ENG^ctrl^* and *ENG^KI^* fibroblasts were incubated with 5 ng/mL TGF-β1 for 24 h, after which *CCN2* (**A**), *SERPINE1* (**B**), *ACTA* (**C**), *FN1* (**D**) and *COL1A1* (**E**) mRNA levels were measured using qPCR. The response to TGF-β stimulation is shown relative to their own unstimulated control. Results were corrected for the housekeeping gene *HPRT*. Summary data are presented as means ± SD from three independent experiments. * *p* < 0.05, ** *p* < 0.01 and *** *p* < 0.001 (Student’s *t*-test after log transformation).

**Figure 5 ijms-24-00646-f005:**
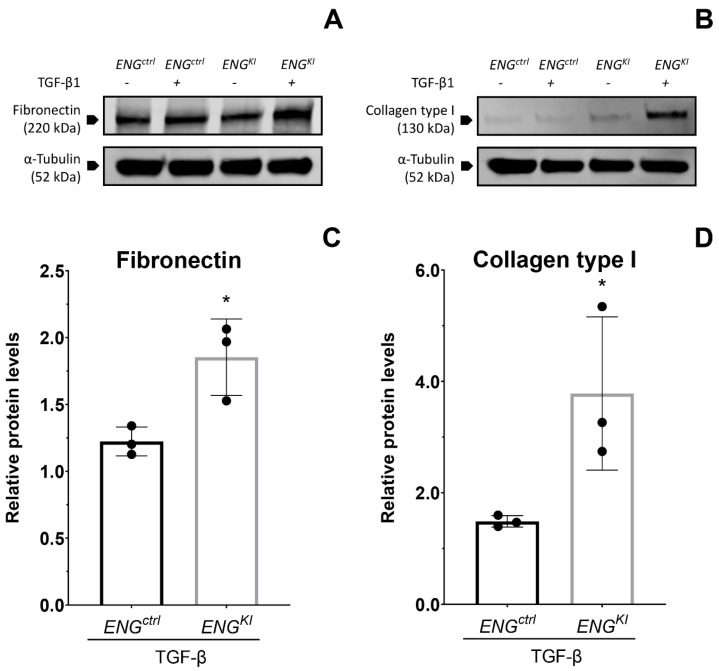
***ENG^K^*^I^ fibroblasts have an increased TGF-β—induced production of fibronectin and collagen type I.** *ENG^ctrl^* and *ENG^KI^* fibroblasts were incubated with (+) and without (-) 5 ng/mL TGF-β1 for 24 h, after which a Western blot for collagen type I (**A**) and fibronectin (**B**) was executed. Relative upregulation of collagen type I (**C**) and fibronectin (**D**) protein levels upon TGF-β stimulation are displayed relative to their own unstimulated control. Protein levels were corrected for the loading control α-Tubulin. Summary data are presented as means ± SD from three independent experiments. * *p* < 0.05 (Student’s *t*-test after log transformation).

**Table 1 ijms-24-00646-t001:** Baseline characteristics of patient cohorts.

Characteristics		(*n* = 170)	CKD (*n* = 163)	ET (*n* = 7)
Male/Female sex		n/n (%/%)	96/67 (58.8/41.2)	3/4 (42.9/57.1)
CAD (*n* = 43)			22/21 (51.2/48.8)	
DN (*n* = 11)			9/2 (81.8/18.2)	
FSGS (*n* = 25)			10/15 (40.0/60.0)	
IgA (*n* = 84)			59/25 (70.2/29.8)	
Age (Years)		Mean ± SD	42.8 ± 20.3	54.8 ± 23.6
CAD			50.1 ± 13.1	
DN			57.7 ± 12.1	
FSGS			33.7 ± 22.1	
IgA			39.9 ± 21.6	
eGFR (mL/min/1.73 m^2^)		Mean ± SD	50.4 ± 28.4	N.A.
CAD			30.0 ± 12.9	
DN			54.2 ± 25.7	
FSGS			75.2 ± 22.0	
IgA			58.4 ± 30.4	
Years after kidney transplantation	CAD	Mean ± SD	6.1 ± 4.9	
Type 1 Diabetes	DN	n (%)	3 (27.3)	

*n* = number, CKD = chronic kidney disease, ET = excluded for transplantation, CAD = chronic allograft dysfunction, DN = diabetic nephropathy, FSGS = focal segmental glomerulosclerosis, IgA = IgA nephropathy, SD = standard deviation, N.A. = not applicable.

**Table 2 ijms-24-00646-t002:** qPCR primers.

Primer Set	Forward	Reverse
*ACTA2*	5′-TTCAATGTCCCAGCCATGTA-3′	5′-GAAGGAATAGCCACGCTCAG-3′
*COL1A1*	5′-GTGCTAAAGGTGCCAATGGT-3′	5′-CTCCTCGCTTTCCTTCCTCT-3′
*CCN2*	5′-CCTGGTCCAGACCACAGAGT-3′	5′-TGGAGATTTTGGGAGTACGG-3′
*ENG*	5′-CACTAGCCAGGTCTCGAAGG-3′	5′-CTGAGGACCAGAAGCACCTC-3′
*FN1*	5′-ACCAACCTACGGATGACTCG-3′	5′-GCTCATCATCTGGCCATTTT-3′
*GFP*	5′-CCCGACACCCACTACCTGAG-3′	5′-GTCCATGCCGAGAGTGATCC-3′
*HPRT1*	5′-AGATGGTCAAGGTCGCAAGC-3′	5′-TCAAGGGCATATCCTACAACAAAC-3′
*SERPINE1*	5′-ACTGGAAAGGCAACATGACC-3′	5′-TGACAGCTGTGGATGAGGAG-3′

## Data Availability

Data sharing not applicable.

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
