# Peer review of "Endoglin Is an Important Mediator in the Final Common Pathway of Chronic Kidney Disease to End-Stage Renal Disease"

_ijms, 2022, doi:10.3390/ijms24010646_

Round 1
Reviewer 1 Report
1. Is endoglin not expressed in TK173 cell line?
2. In this research, the research indicated that endoglin overexpression increases the TGF-β-induced profibrotic response in TK173 cell line. Howerver, this research did not investigate the reduction of endoglin in the TGF-β-induced profibrotic response. Why not construct a TK173 cell line with endoglin deleted?
3. Some tags in the images use full names and some abbreviations (for example, ENG and Endoglin). Please unify the standard.
4. To investigate the effect of the increased endoglin expression on the TGF-β induced 177 fibrotic response, mRNA expression of ACTA2, CTGF, SERPINE-1, FN and Col1A1 was measured using quantitative real-time PCR (qPCR). Why were the protein expressions levels of these factors not measured.
Reviewer 2 Report
Good research. CKD is a huge public health challenge and identifying potential markers like Endoglin is very interesting.
The analysis is statistically sound. However if the reason why arcsine transformation is used for ANOVA for analyzing endoglin expression in various CKDs instead of logit transformation.
